# Plant Regeneration via Adventitious Shoot Formation from Immature Zygotic Embryo Explants of Camelina

**DOI:** 10.3390/plants13040465

**Published:** 2024-02-06

**Authors:** Barno Ruzimurodovna Rezaeva, Twan Rutten, Carola Bollmann, Stefan Ortleb, Michael Melzer, Jochen Kumlehn

**Affiliations:** 1Plant Reproductive Biology, Leibniz Institute for Plant Genetics and Crop Plant Research (IPK), 06466 Seeland, Germany; rezaeva@ipk-gatersleben.de (B.R.R.); bollmann@ipk-gatersleben.de (C.B.); 2Structural Cell Biology, Leibniz Institute for Plant Genetics and Crop Plant Research (IPK), 06466 Seeland, Germany; rutten@ipk-gatersleben.de (T.R.); melzer@ipk-gatersleben.de (M.M.); 3Assimilate Allocation and NMR, Leibniz Institute for Plant Genetics and Crop Plant Research (IPK), 06466 Seeland, Germany; ortleb@ipk-gatersleben.de

**Keywords:** Auxin, 6-benzylaminopurine, *Camelina sativa*, cytokinins, histology, hypocotyl, indole-3-acetic acid, protuberance, wound response

## Abstract

Camelina is an oil seed crop that is enjoying increasing interest because it has a particularly valuable fatty acid profile, is modest regarding its water and nutrient requirements, and is comparatively resilient to abiotic and biotic stress factors. The regeneration of plants from cells accessible to genetic manipulation is an essential prerequisite for the generation of genetically engineered plants, be it by transgenesis or genome editing. Here, immature embryos were used on the assumption that their incomplete differentiation was associated with totipotency. In culture, regenerative structures appeared adventitiously at the embryos’ hypocotyls. For this, the application of auxin- or cytokinin-type growth regulators was essential. The formation of regenerative structures was most efficient when indole-3-acetic acid was added to the induction medium at 1 mg/L, zygotic embryos of the medium walking stick stage were used, and their hypocotyls were stimulated by pricking to a wound response. Histological examinations revealed that the formation of adventitious shoots was initiated by locally activated cell division and proliferation in the epidermis and the outer cortex of the hypocotyl. While the regeneration of plants was established in principle using the experimental line Cam139, the method proved to be similarly applicable to the current cultivar Ligena, and hence it constitutes a vital basis for future genetic engineering approaches.

## 1. Introduction

Camelina [*Camelina sativa* (L.) Crantz] is an oilseed crop plant belonging to the family Brassicaceae. The first traces of its cultivation date back nearly 4000 years [1]. Camelina remained an important oilseed crop until the 1950s, when it was largely replaced with other oil-producing plants, including rapeseed (*Brassica napus* L.) [2]. In recent decades, however, we have been witnessing a renaissance in the use of camelina. Not only can camelina be used as an intercrop thanks to its short growth period, but it also has low water and nutrient requirements and is comparatively resilient against various adverse abiotic stresses as well as pathogens and pests. The high oil content of its seeds, along with a unique composition of fatty acids, renders this crop particularly useful for food and industrial applications [3]. The essential linoleic and α-linolenic acids make up more than 50% of the total fatty acid content. Especially the high proportion of the polyunsaturated omega-3 α-linolenic acid (30–43%) is only surpassed in linseed oil [4]. The oil content of camelina seeds has been reported to vary between 30 and 49% depending on genotype, environmental factors, and agronomic practice [5]. For similar reasons, the yield of oil per hectare can range by almost one order of magnitude [6]. Nowadays, the quality characteristics of camelina are subject to both conventional breeding and biotechnological approaches, and camelina has become one of the very first crop plants for which genome-edited lines have been tested in the field [7].

The in vitro regeneration of plants is a basic prerequisite for the development and application of various biotechnological methods. Camelina plants have been regenerated via adventitious shoot formation from seedling-derived explants such as hypocotyls, cotyledon petioles, and true leaves [8,9]. Immature zygotic embryos dissected from developing seeds proved to be particularly useful for in vitro plant regeneration via adventitious shoot formation or somatic embryogenesis in a variety of plant species. In cereals, plant regeneration from immature zygotic embryos has been broadly used [10]. In addition, this principle was also established in dicotyledons and, more specifically, in species where less satisfactory results had been obtained by other approaches. Examples are sunflower [11], pepper [12], soybean [13], grapevine [14], woody species [15,16], and some brassicas [17,18,19].

As a facile alternative for genetic engineering, the tissue culture-independent principle of dipping the flowers into *Agrobacterium* suspension has been applied in camelina [20,21]. However, as in some other crop species, genetic engineering of camelina has remained limited owing to low or inconsistent efficiency and high genotype dependence, irrespective of the employed methods of DNA transfer and plant regeneration.

The aim of this study was to establish a novel method of in vitro regeneration based on the formation of adventitious shoots on immature zygotic embryo explants, which has the potential to be applicable to a wide range of camelina accessions. The rationale behind this approach is that immature embryos are more likely to contain totipotent cells as compared with explants taken from mature embryos or plants. As a result, a robust method was established that rests on neoplastic development at the hypocotyl of cultivated immature zygotic embryos. The resulting protuberances further develop via the emergence of leafy structures into adventitious shoots and, after rooting, give rise to plants. This method was largely established in the experimental line Cam139 and then adopted in the current cultivar, Ligena. In addition, to gain a deeper understanding of the underlying developmental process, the adventitious formation of regenerative structures was histologically traced back to the initiation of locally enhanced cell proliferation in the outermost layers of the hypocotyl. Our results constitute a promising precondition for the establishment of robust methods of genetic engineering, including genome editing in camelina.

## 2. Results

### 2.1. Effect of Growth Regulators on the Adventitious Formation of Regenerative Structures

To establish whether immature zygotic embryos can be stimulated by externally added auxin or cytokinin to form regenerative structures, embryos of camelina accession Cam139 were cultured on media containing 0.2 to 2 mg/L of the auxin indole-3-acetic acid (IAA) or of the synthetic cytokinin analog 6-benzylaminopurine (BAP). Figure 1A,B shows immature siliques, isolated immature seeds, and dissected embryos used for this study. For both growth regulators, the concentration of 1 mg/L proved to be the most effective (Figure 2). After three to four days of culture in the presence of 1 mg/L BAP, green outgrowths formed at the lower part of the hypocotyl (Figure 1C), which eventually assumed a characteristic cylindric shape (Figure 1D). By contrast, in the presence of 1 mg/L IAA, outgrowths only occurred in the upper part of the hypocotyl near the cotyledons and did not become visible to the naked eye until after one week of culture (Figure 1E). In comparison to BAP, the IAA-induced structures were more leaf-shaped (Figure 1E,F). These major characteristics were highly consistent among explants cultivated under the same conditions and even across the three size classes of cultivated embryos. The IAA-induced developmental pattern was also observed when IAA was replaced by the auxin-type growth regulators naphtylacetic acid (NAA), 2,4-dichlorophenoxy acid (2,4-D), or Dicamba. Nevertheless, none of these substances improved the overall formation of leafy structures (Figure 3).

### 2.2. Impact of the Developmental Stage of Embryos on Their Response in Culture

To investigate the response of immature embryos at different developmental stages, early, medium, and late walking stick embryos were compared. Across all stages, embryos proved capable of adventitiously forming leafy structures within 2 weeks of cultivation. The highest efficiency was obtained with embryos of the medium walking stick stage, irrespective of the growth regulator used (Figure 4). The results were consistently better when IAA was used instead of BAP. Remarkably, some of the late walking stick stage embryos elongated their apical part without forming any adventitious structures at the hypocotyl. This was observed on both IAA- and BAP-containing culture media.

### 2.3. Development of Immature Embryo Explants of the Current Camelina Cultivar Ligena

After hypocotyl-derived formation of leafy structures had been established for accession Cam139, the method was also applied to the randomly chosen cv. Ligena to test and demonstrate its applicability to agronomically used germplasm. On induction medium supplemented with 1 mg/L BAP, explants of cv. Ligena largely behaved like those of Cam139, including the emergence of mainly cylindrical structures at the lower part of the hypocotyl. On induction medium containing 1 mg/L IAA, however, the cv. Ligena explants responded differently, developing two parallel rows of leafy structures emerging along the whole length of the hypocotyl (compare Figure 1 and Figure 5). Of note, this phenomenon was largely confined to the hypocotyl area in contact with the culture medium.

### 2.4. Conversion of Hypocotyl-Derived Leafy Structures to Shoots and Plants

At the end of four weeks of in vitro cultivation, some leafy structures had converted into shoots that featured a rosette of true leaves (Figure 6A). Shoots individually cut off from hypocotyl explants grown on IAA-containing media and transferred to rooting medium formed roots at their basal ends (Figure 6B) with 85.3% (81 plantlets from 95 shoots) and 84.2% (101 plantlets from 120 shoots) efficiency in lines Cam139 and cv. Ligena, respectively. In contrast, shoots that had emerged on BAP-containing shooting medium were less capable of forming roots, that is, only 26% (23 plantlets from 88 shoots) and 24% (22 plantlets from 91 shoots) rooting efficiencies were observed under this condition in lines Cam139 and cv. Ligena, respectively.

In both genotypes tested, the whole tissue culture process took eight weeks to develop rooted shoots ready for transfer to soil substrate. Almost all plantlets formed in vitro were successfully acclimated to glasshouse conditions (Figure 6C). The resultant plants grow vigorously, producing flowers and, after pollination, seed-filled silicles.

### 2.5. Effect of Mechanical Wounding on Adventitious Shoot Formation

The most efficient shoot formation was achieved in cv. Ligena, when the explants had been wounded by pricking their hypocotyl and cultured on the induction medium containing IAA. In variety Cam139, however, both ballistic wounding and pricking improved shoot formation to a similar extent as compared with control conditions (Figure 7).

Callus formation was only rarely observed in the present investigation, whereas embryo-like structures occasionally arose from the zygotic embryo explants without preceding callus formation, which was the case on both IAA- and BAP-containing media and in both genotypes used (Figure 5C). These structures had a pale-green color and a smooth surface, and occurred more often upon wounding. However, somatic embryogenesis only marginally contributed to plant regeneration.

### 2.6. Structural Analysis of the Formation of Leafy Structures

Cultured embryos of Cam139 and cv. Ligena were microscopically examined to identify the regions within the hypocotyl where the formation of leafy structures is initiated. Transverse sections reveal a central pro-vascular structure surrounded by a cortex comprising ca. five cell layers, and concluded with a single-layered epidermis (Figure 8A). In freshly isolated immature zygotic embryos, epidermal and cortex cells display a dense cytoplasm filled with numerous lipid droplets and starch granules, interspaced with multiple small vacuoles (Figure 8A–D). When visible, nuclei had a central position.

After three days of cultivation without exogenously added growth regulators, a major increase in vacuolar size causes a strong expansion of epidermal and cortical cells (Figure 8E,F). Despite the cytoplasm becoming less dense, aggregations of starch granules remain common, yet lipid droplets have become exceedingly rare (Figure 8G,H). Due to the overall increase in cell size, nuclei are rarely visible in singular sections. The epidermal cell layer occasionally showed anticlinal cell divisions (Figure 8F,G).

In contrast to the control conditions, hypocotyls cultivated for three days on BAP-containing medium showed a high frequency of periclinal and anticlinal cell divisions that were most common in the epidermis and subepidermal cortex (Figure 8I–K). Neoplastic development became microscopically visible at the surface of hypocotyls after four days of culture (Figure 1C). It can thus be regarded as a direct result of the local increase in mitotic activity. While dividing cells appeared to be enriched in lipid droplets, starch granules were predominantly present in those cells not involved in cell division (Figure 8K,L). A similar pattern was observed in hypocotyls cultured for four days on IAA-containing induction medium (Figure 8M–P). Under this condition too, mitotically active cells were characterized by a comparatively dense cytoplasm. In Cam139, the intensity of cell proliferation appeared to be higher on IAA-containing than on BAP-containing induction medium, even though in the latter case, this process started slightly earlier.

To enhance our understanding of the relationship between immature embryos and leafy outgrowths, 3-D reconstructions were made using serially sectioned cv. Ligena hypocotyls after two weeks on IAA- or BAP-containing media (Figure 9, Appendix A). In both cases, the straight, singular, unbranched veins of the leafy structures seem to originate from the outermost cell layers of the hypocotyl, and definitively do not contact the main vein of the hypocotyl. In contrast to this, the veins of partially removed cotyledons and true leaves display a reticulate pattern directly linked to the main vein of the shoot (Figure 9, Appendix A).

In summary, the cultivation of immature embryos on BAP- or IAA-containing media caused strong mitotic activity in the epidermis and outer cortical layers of the hypocotyl, identifying these regions as the origin for the formation of leafy structures and their further development into shoots and regenerated plants.

## 3. Discussion

The aim of this study was to tap the expectedly high totipotency of cells in zygotic immature camelina embryos for the establishment of a new, robust regeneration system. The method to be established should not only be applicable to particularly suitable experimental lines but also to elite breeding material. The in vitro cultivation of such embryos under the conditions established in this study led to local protuberances at the surface of the embryonic hypocotyl. From these tissue areas with intensified cell division, adventitious shoots were formed, some of which converted into normally developing plants.

In grasses, as compared with many dicotyledonous species, it is, if not impossible, at least more challenging to redirect the development of vegetative tissue to the formation of regenerative structures such as adventitious shoots or somatic embryos [22]. Therefore, immature zygotic embryos constitute the standard explant for in vitro regeneration in these species. However, this principle has not yet been widely applied to dicotyledonous plant species, with one of the exceptions being Chinese cabbage, where the efficiency of regenerative structure formation using immature embryos was indeed higher than with other explants [23]. The present investigation was the first to utilize this principle in the genus *Camelina*.

The dependence of the in vitro development of explants on their developmental stage is a well-known phenomenon. In the present study, immature embryos of the medium walking stick stage proved to be best suited for the formation of adventitious leafy structures (Figure 4). While younger-stage embryos appeared to be only marginally more sensitive to the dissection procedure and their changing environment as compared with medium-size ones, embryos of the late walking stick stage exhibited a statistically significantly reduced formation of regenerative structures, especially when cultivated in the presence of IAA (Figure 4). In this respect, our results are on par with earlier observations for cabbage, cauliflower, and kohlrabi [18,19] and support the hypothesis that, due to advanced cellular differentiation, embryos of the later developmental stage feature a decreasing totipotency of their cells.

Based upon histological preparations, the adventitious structures originated from the outer cell layers of the hypocotyl, i.e., epidermal cells, as well as subepidermal cells of the outer cortex. During the formation of the protuberances, the original order of the cell layers within the hypocotyl, largely based on anticlinal cell divisions, was abolished due to a strong increase in periclinal cell divisions. The simultaneous division of several juxtaposed cells and the broad base of the protuberances speak for a multicellular origin of regeneration, which may be relevant for the later employment of the established culture system for genetic engineering approaches. Similar cell division patterns were also observed at the beginning of the formation of regenerative structures in sunflower [24] and Chinese tallow trees [25], where an increasing proportion of periclinal cell divisions either contributed to internally unstructured protuberances or led to the formation of somatic embryos. By contrast, somatic embryos developed too rarely under the conditions described here to enable us to clarify their cellular origin with feasible effort using histological preparations.

An interesting observation in the present study was that the preferential site of adventitious emergence of leafy structures was dependent on the exogenously supplied growth regulator. When the cytokinin analog BAP was administered as the sole growth regulator, the leafy structures emerged preferentially at the basal end of the hypocotyl, a phenomenon that was also observed in sunflower and Chinese tallow trees [24,25]. In the presence of the auxin IAA, however, the leafy structures were formed either at the upper end of the hypocotyl (in Cam139) or along its entire length (in cv. Ligena).

Cultivation of immature embryos on induction medium without growth regulators overall led to a strong cell expansion driven by vacuole enlargement in cortex and epidermis of the hypocotyl but had no obvious effect on cell division. The formation of adventitious structures was observed exclusively when the culture medium was supplemented with auxin- or cytokinin-type growth regulators. It was somewhat surprising that the auxin analogs NAA, 2,4-D, and Dicamba did not lead to a more intensive formation of adventitious structures compared to IAA (Figure 3) that is known to be comparatively unstable in aqueous solutions. Further investigations showed that the formation of regenerative structures occurred most efficiently at a concentration of 1 mg/L of IAA or BAP (Figure 2). In some previous studies, however, adventitious shoot formation on cultured immature embryos in the *Brassicaceae* cabbage, kohlrabi, and cauliflower was most efficient without supplemented growth regulators [18,19]. Apparently, in these species, the dissection of the immature embryo from the seed and the altered conditions associated with the cultivation in vitro were sufficient to trigger the developmental changes.

In the literature, the use of phytohormones or their synthetic analogs to induce regenerative structures on immature embryos of dicotyledonous species ranges from IAA (*Datura innoxia* [26]), NAA (rapeseed [27]), 2,4-D (Arabidopsis [28], pepper [12], *Brassica rapa* [23], sunflower [11,13]), BAP (sunflower [24], neem [15]) to various combinations of auxins and cytokinins. Adventitious shoots and somatic embryos have been obtained from the cotyledons of immature embryos of, e.g., rapeseed [27] and Arabidopsis [29] in addition to the hypocotyl of *Datura innoxia* [26], sunflower [13], kohlrabi [18], cabbage, and cauliflower [19].

Serial section analysis of developing explants revealed that the vascular bundles of the adventitious leafy structures are largely straight, singular, and without branches. Irrespective of the use of either IAA or BAP for the induction of adventitious structures, the vasculature appears to originate within the protuberances, and definitively does not extend towards the main vasculature of the hypocotyl (Figure 9). By contrast, the connection between the vasculatures of regenerative structures and explants was previously reported to be associated with adventitious shoot formation, for instance in *Hypericum perforatum* and Arabidopsis [29,30]. Based on such observations, the prevailing view in the field has emerged that the vascular connection with the explant is a suitable distinguishing feature between adventitious shoot formation and somatic embryogenesis [31]. However, the microscopic investigations presented in our study suggest that such a generalization is no longer justifiable. Irrespective of this new insight, the unipolar development of adventitious shoots and the bipolar development of the apical shoot meristem and apical root meristem of somatic embryos remain unambiguous differentiation criteria.

In this study, we tested the effect of mechanical wounding by ballistic particles or pricking with an injection needle on the formation of regenerative structures. Both approaches promoted the emergence of adventitious structures as compared with non-wounded hypocotyls grown under otherwise identical culture conditions (Figure 6). The current hypothesis is that wounding causes the transfer of endogenous auxin to the wounded sites and hence an accumulation there [32]. In this respect, results in line with ours were previously reported in, e.g., soybean and Chinese tallow trees [13,25].

The achievements of the present investigation are difficult to compare with previous studies in which immature embryos were used, since the overall efficiency of the established methods was rarely provided. In one of the exceptions to this, Kraut et al. [17], working on Arabidopsis, reported an average of about one regenerated plant per immature embryo explant, which is comparable to our results. A comparison with previous studies on the in vitro culture of other sorts of camelina explants also proves to be difficult. Pollard et al. [33] cultivated immature embryos of camelina cv. Sunesson to investigate the contents and fluxes of metabolites. However, the development of the cultured embryos was beyond the scope of their study. In contrast, Tattersall and Millam [8] and Yemets et al. [9], aiming to achieve multiple shoot formation from seedling explants, used segments from leaves of a Danish camelina accession and from hypocotyls or petioles of the Ukrainian camelina cultivars Peremozhets and Mirazh, respectively. In both studies, adventitious shoots developed from calluses that had initially formed. Unfortunately, only anecdotal information was provided in the two studies [8,9] on shoot formation and the conversion of the shoots into plants. In contrast to the previous reports on camelina, the regeneration of shoots and plants in the method developed by us usually takes place without an intermediate formation of callus, which can be of great advantage with regard to the genetic and epigenetic stability of the resulting plants. Also worth mentioning in this context is that comparatively high numbers of leafy structures can be obtained by the method established here, so that the conversion of these structures into meristem-bearing shoots and plants may be further improved.

While the dipping method is exclusively suitable for *Agrobacterium*-mediated DNA transfer, the peripheral localization of the founder cells of plant regeneration in the method established here suggests that they may be well accessible for the ballistic transfer of transgene-coding plasmids or other reagents such as pre-produced ribonucleoprotein complexes (RNPs) of Cas-endonuclease and guide RNA. This opens additional opportunities, especially for genome editing, where the delivery of RNPs as well as DNA repair templates at high dosages is of great interest for homologous recombination-mediated precise editing [34].

Taken together, a robust method of rapid plant regeneration was established using immature zygotic embryos of camelina, including the current cultivar Ligena, for which no regeneration or transformation method had previously been available. Under further consideration, the fact that the regenerated plants have their cellular origin in the outermost cell layers of the explants represents an extraordinarily favorable starting point for the intended application of this regeneration principle towards the development of methods of transgenesis and genome editing in this crop.

## 4. Materials and Methods

Experiments were performed using camelina accession Cam139 (provided by the Genebank of the Leibniz Institute of Plant Genetics and Crop Plant Research (IPK) Gatersleben, Gatersleben, Germany) and camelina cultivar Ligena (provided by Dr. Dieter Stelling from Deutsche Saatveredelung, Lippstadt-Bremen, Germany). Plants were grown in a greenhouse under natural daylight supplemented with 16 h illumination by sodium high-pressure lamps, providing an additional photon flux density of 300 to 500 μmol m^−2^ s^−1^ depending on the plants’ vertical distance and individual position. The day/night temperature regime was adjusted to 20/18 °C, and the relative humidity to 65%. To obtain zygotic embryos, silicles were harvested 12–14 days after pollination (Figure 1) without distinguishing any different fruit size classes. After surface sterilization for 20 min with commercial bleach diluted to 4% NaOCl and supplemented with 0.1% Tween 20, silicles were rinsed three times with distilled water and stored at 4 °C overnight, which facilitates the dissection of the immature seeds. Isolation of immature zygotic embryos and all subsequent processing steps were performed under aseptic conditions in a laminar airflow workstation.

### 4.1. Culture of Immature Zygotic Embryo Explants

Zygotic embryos were dissected from immature seeds using a Zeiss Stemi 2000 binocular (Carl Zeiss Microscopy GmbH, Jena, Germany), with the embryos being sorted according to their individual length (for embryo size classes, see next paragraph) and cultivated on various culture media to establish conditions for the formation of adventitious regenerative structures. Per 90-mm plastic plate, 25 mL of induction medium was used, which contained, if not stated otherwise, ammonium-free MS basal minerals (Duchefa Biochemie, Haarlem, The Netherlands), 5 mM NH_4_NO_3_, B5 vitamins (Sigma-Aldrich, St. Louis, MO, USA), 3% (*w*/*v*) sucrose, and 0.4% Phytagel (*w*/*v*) (Sigma-Aldrich) for solidification, with the pH being adjusted to 5.7 with KOH. Growth regulators were added, as specified below. A 2× stock solution of induction medium without Phytagel was filter-sterilized (nonpyrogenic, sterile polystyrene filter, Corning, New York, NY, USA), while a 2× stock solution of Phytagel was autoclaved at 120 °C for 20 min. After warming the former and cooling the latter to about 50 °C, the two solutions were mixed 1:1 and distributed to the culture plates for solidification.

The plant growth regulators Dicamba, α-naphthaleneacetic acid (NAA), indole-3-acetic acid (IAA), 2,4-dichlorophenoxyacetic acid (2,4-D), or 6-benzylaminopurine (BAP) (all from Duchefa Biochemie Co., Haarlem, The Netherlands) were added to the medium as filter-sterilized, 1 mg/L stock solutions. After testing IAA and BAP at concentrations of 0.2, 0.5, 1.0, and 2.0 mg/L, final experiments were carried out using either IAA or BAP at a concentration of 1 mg/L. The developmental stages of regular embryonic development were specified according to Feeney et al. [35]. To identify the most suitable developmental stage for adventitious shoot formation, early (>0.7–1.2 mm in length), medium (>1.2–1.7 mm), and late walking stick (>1.7–2.1 mm) zygotic embryos were tested. Ten dissected embryos were cultivated in a 9-cm culture dish. Cultures were kept overnight in the dark at 24 °C before being incubated at 25 °C with a 16 h-photoperiod using white fluorescent lamps with a 60 µmol m^−2^ s^−1^ photon flux density. After 2 weeks of culture, the number of leafy structures formed per immature zygotic embryo was counted.

### 4.2. Adventitious Shoot Formation and Plant Regeneration

After two weeks of cultivation, the cotyledon and root parts were removed, and the remaining hypocotyls carrying the emerging leafy structures were placed horizontally onto the shooting medium. This medium had the same composition as the induction medium in the case of explants cultivated in the presence of BAP, whereas for explants from the IAA-containing induction medium, sucrose was reduced to 2% (*w*/*v*) and IAA to 0.1 mg/L.

After two weeks on shooting medium, resultant shoots were transferred onto rooting medium consisting of half-strength ammonium-free MS minerals (Duchefa Biochemie, Haarlem, The Netherlands), 2.5 mM NH_4_NO_3_, 1% (*w*/*v*) sucrose, and 0.1 mg/L IAA. After another four weeks of cultivation, rooted shoots were counted and transferred to pots with soil substrate to grow them under the conditions used for the donor plants. Images of explants and their development in vitro were documented using a KEYENCE VHX-5000 microscope (Keyence Germany GmbH, Neu-Isenburg, Germany).

### 4.3. Mechanical Wounding of Hypocotyls

To examine whether the formation of adventitious shoots can be enhanced by mechanical wounding, medium walking stick stage embryos were first cultivated for 72 h on induction medium containing either 1 mg/L BAP or IAA. In this experiment, 20 explants were used per culture dish. The hypocotyls were then carefully pricked at 3 to 5 positions with an injection needle (0.3 mm × 25 mm, BD PrecisionGlide^TM^, Fort Pierce, FL, USA), followed by cultivation under the same conditions for another two weeks. For comparison, a second series of immature zygotic embryos were wounded by accelerated particles using a PDS-1000/He device equipped with a hepta-adapter (Bio-Rad Laboratories, Hercules, CA, USA). For this, 30 mg of gold particles (ca. 0.6 μm in diameter; Bio-Rad, order no. 1652262) were washed twice with 200 µL of H_2_O in a 0.5-mL microcentrifuge tube by briefly vortexing. After the supernatant was removed, the particles were resuspended in 60 µL of 100% ethanol. Each macro-carrier (Bio-Rad, order no. 1652335) was loaded with 5 µL particle suspension. Prior to the experiment, immature zygotic embryos were transferred from the induction medium to the osmotic medium (ammonium-free MS minerals, B5 vitamins (Sigma-Aldrich), 0.35 M mannitol, and 0.4% Phytagel) and incubated for four to six hours at 24 °C in the dark before use. The particle gun was employed with a vacuum of 27 inches and 1100-psi rupture discs (Bio-Rad, order no. 1652329). Per shot, 20 immature zygotic embryos were used. After this ballistic treatment, the embryos remained on the osmotic medium overnight in the dark at 24 °C. The next day, the explants were transferred back to induction medium and incubated in the dark at 24 °C for another two weeks. Wounded embryos were compared with non-wounded, immature embryos serving as controls.

### 4.4. Statstical Analyses

Using the culture dishes as experimental repetitions, the experimental datasets were tested for normal (Gaussian) distribution and equal variance. Provided these tests were passed, an analysis of variance (ANOVA) was conducted, followed by all pairwise multiple comparisons of the treatments using the Tukey test. By contrast, datasets that did not pass both the normal distribution and equal variance tests were subjected to the non-parametric Kruskal-Wallis analysis of variance on ranks, followed by all pairwise multiple comparisons of the treatments using the Student-Newman-Keuls method. All statistical analyses were performed using the SigmaPlot 14.0 software package (Inpixon GmbH, Düsseldorf, Germany).

### 4.5. Microscopic Analyses

For histological analysis, medium-size walking stick zygotic embryos were used either freshly prepared or after 3 days of culture on BAP- or 4 days on IAA-containing media. The longer cultivation in the presence of IAA was used to compensate for the apparently faster development with BAP. For comparison, embryos were cultivated for four days on growth regulator-free medium.

After removal of the cotyledons’ node and the root, hypocotyl segments were fixed with 2.0% (*v*/*v*) glutaraldehyde and 2.0% (*w*/*v*) formaldehyde in 50 mM cacodylate buffer (pH 7.2) for 2 h at room temperature. After three rinses with buffer and one rinse with aqua dest., samples were post-fixed with 1% (*w*/*v*) aqueous OsO_4_ for 30 min. After another three washing steps with aqua dest., samples were dehydrated in an ascending ethanol series of 30% to 100%. After an additional incubation with 100% propylene oxide followed by a stepwise infiltration with Spurr’s resin (Plano GmbH, Marburg, Germany), fully infiltrated samples were transferred into embedding forms (BEEM capsules, Plano GmbH, Wetzlar, Germany) and polymerized overnight in a heating cabinet at 70 °C. Resin-embedded samples were trimmed with a Leica EM TRIM2 (Leica Microsystems, Wetzlar, Germany) device. Sections of 2-µm thickness were cut using a Leica Ultracut Ultramicrotome (Leica Microsystems, Wetzlar, Germany) and stained with 1% (*w*/*v*) methylene blue/1% (*w*/*v*) Azur II in 1% (*w*/*v*) aqueous borax. Stained sections were examined in a Zeiss Axio Imager 2 light microscope (Carl Zeiss GmbH, Jena, Germany).

For 3-D reconstruction of explants with adventitious leafy structures, samples were serially sliced to generate 2-µm-thick sections, of which those positioned at 10-µm intervals were collected. Sections were transferred onto 20-µL droplets of 1:400 diluted Crystal Violet (2% stock) on poly-l-lysine-coated slides and placed on a heating plate set at 90 °C to simultaneously stain and bake the sections. Recordings were made with a Zeiss Axioscan 7 slide reader (Carl Zeiss, Oberkochen, Germany) using a 10× NA 0.45 objective. Serial recordings were stacked and aligned using the open-source Fiji image processing software (version 1.54c, National Institutes of Health and Laboratory for Optical and Computational Instrumentation, University of Wisconsin, USA). For subsequent segmentation and 3-D reconstruction, Amira software (version 5.6, Thermo Fischer, Karlsruhe, Germany) was used.

Ultrastructure analyses by transmission electron microscopy of resin-embedded samples were performed as described previously [36].

## Figures and Tables

**Figure 1 plants-13-00465-f001:**
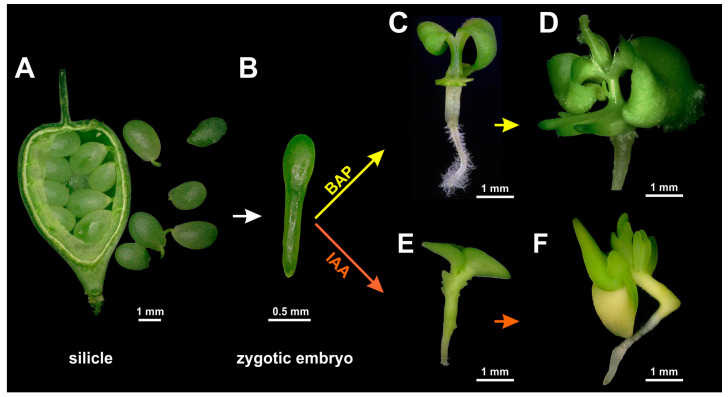
Formation of leafy structures at the hypocotyl of immature zygotic embryos (IZEs) of camelina accession Cam139 was grown on induction medium supplemented with indole-3-acetic acid (IAA) or 6-benzylaminopurine (BAP). (**A**) Freshly collected silicle with immature seeds, (**B**) immature zygotic embryo at the medium walking stick stage, (**C**) leafy structures originating from the lower part of the hypocotyl upon IZE cultivation on BAP-containing induction medium, (**D**) elongation of leafy structures, (**E**,**F**) initiation of leafy structures at the upper part of the hypocotyl after one (**E**) and two (**F**) weeks of culture on IAA-containing induction medium.

**Figure 2 plants-13-00465-f002:**
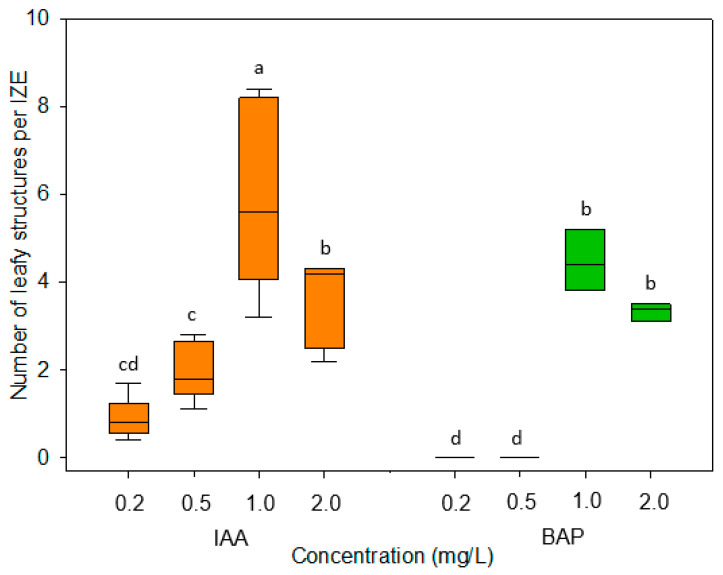
Number of leafy structures formed from immature zygotic embryos (IZEs) of camelina cultivated on induction media supplemented with various concentrations of IAA and BAP. A total of 90 medium-size IZEs were cultivated per treatment, with 10 IZEs per culture dish. Data were recorded after 2 weeks of culture. The boxplots represent the median (central horizontal line) and quartiles of data points, with the whiskers indicating minimum and maximum values. The data were subjected to Kruskal-Wallis analysis of variance on ranks, followed by all pairwise multiple comparisons of treatments using the Student-Newman-Keuls method. Exclusively different letters above any two compared boxplots denote a significant difference between the respective treatments (*p* ≤ 0.05).

**Figure 3 plants-13-00465-f003:**
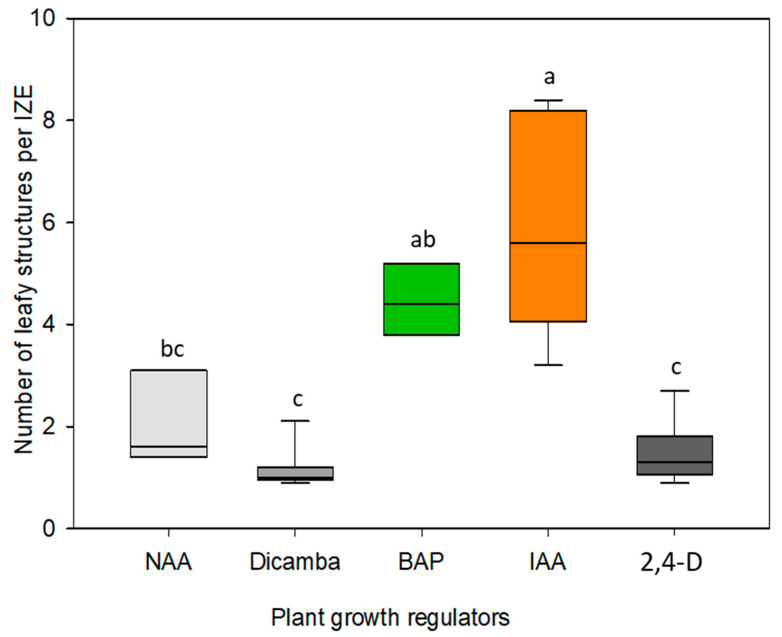
Number of leafy structures formed from immature zygotic embryos (IZEs) of camelina cultivated on induction media supplemented with various growth regulators using 1 mg/L. A total of 90 medium-size IZEs were cultivated per treatment, with 10 IZEs per culture dish. Data were recorded after 2 weeks of culture. The boxplots represent median (central horizontal line) and quartiles of data points, with the whiskers indicating minimum and maximum values. Data were subjected to Kruskal-Wallis analysis of variance on ranks, followed by all pairwise multiple comparisons of treatments using the Student-Newman-Keuls method. Exclusively different letters above any two compared boxplots denote a significant difference between the respective treatments (*p* ≤ 0.05).

**Figure 4 plants-13-00465-f004:**
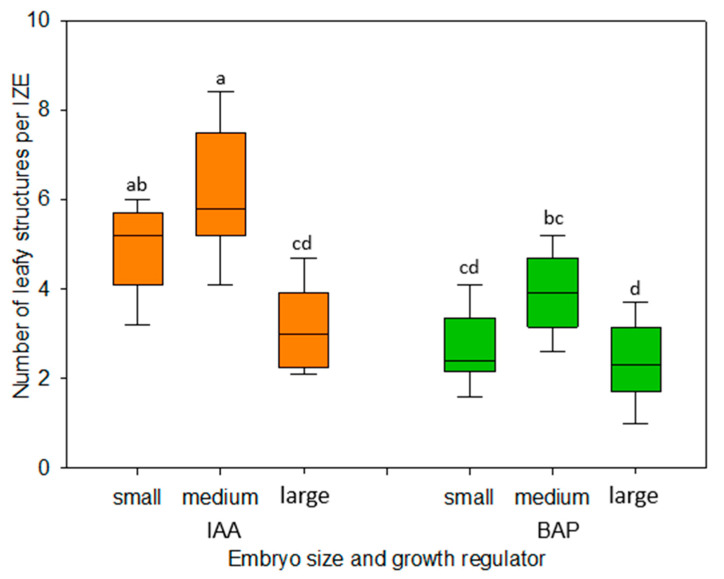
Number of leafy structures formed per immature zygotic embryo (IZE) isolated at 3 developmental stages (early, medium, and late walking stick embryos) and cultivated on induction medium supplemented with either IAA or BAP using 1 mg/L. A total of 90 IZEs were cultivated per treatment, with 10 IZEs per culture dish. Data were recorded after 2 weeks of culture. The boxplots represent median (central horizontal line) and quartiles of data points, with the whiskers indicating minimum and maximum values. Data were subjected to standard analysis of variance, followed by all pairwise multiple comparisons of treatments using the Tukey test. Exclusively different letters above any two compared boxplots denote a significant difference between the respective treatments (*p* ≤ 0.05).

**Figure 5 plants-13-00465-f005:**
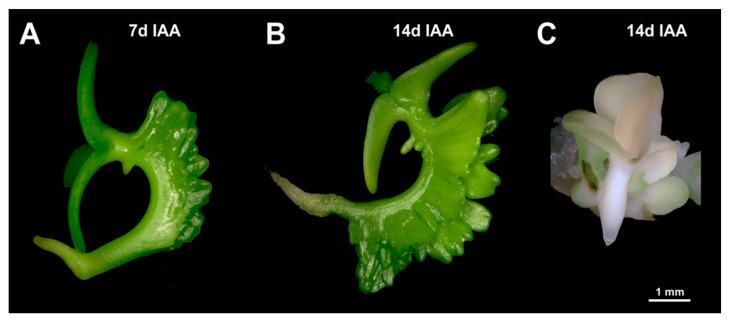
Leafy structures formed along the entire hypocotyl of immature zygotic embryos of camelina cv. Ligena after one (**A**) and two weeks (**B**) of cultivation on induction medium supplemented with IAA. (**C**) Embryo-like structure seen in the foreground that emerged without intermediate callus formation from an immature zygotic embryo after two weeks of cultivation. The given size bar refers to A, B, and C.

**Figure 6 plants-13-00465-f006:**
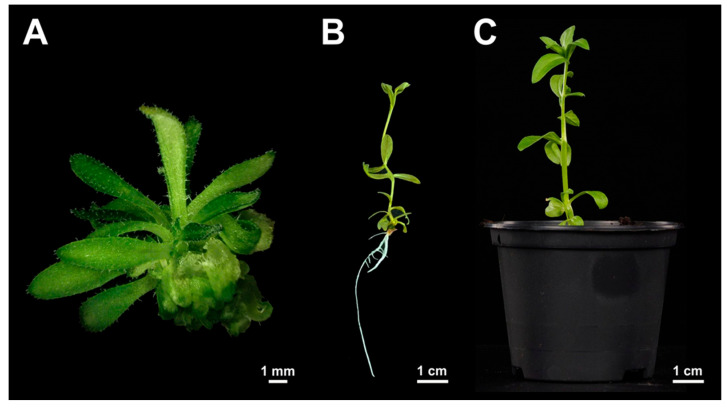
Conversion of adventitious shoots into plants. (**A**) Adventitious shoots stemming from leafy structures after four weeks of culture of zygotic embryo explants, (**B**) individually cultivated adventitious shoots exhibiting root formation after 4 weeks of culture on rooting medium, (**C**) adventitious shoot-derived plantlets established in soil after 10 weeks of culture.

**Figure 7 plants-13-00465-f007:**
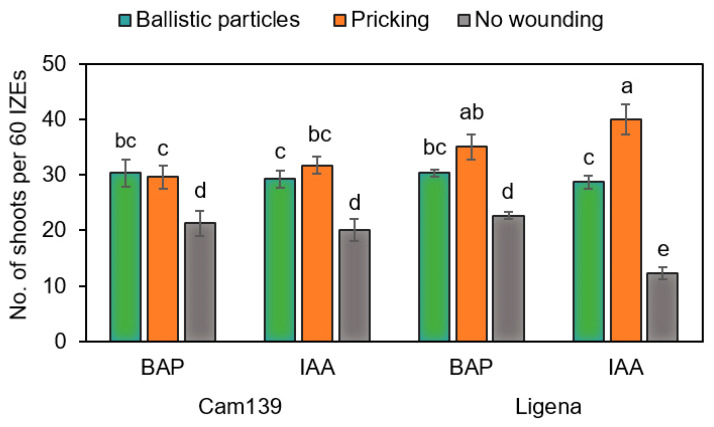
Effect of wounding and genotype on the frequency of adventitious shoot formation from medium-size immature zygotic embryos (IZEs) on culture media containing 1 mg/L IAA or BAP. In total, 60 IZEs were cultivated per variety and treated with 20 embryos per culture dish. Data were recorded after 10 weeks of culture. The bars represent standard deviations. Data were subjected to standard analysis of variance, followed by all pairwise multiple comparisons of treatments using the Tukey test. Exclusively different letters above any two compared columns denote a significant difference between the respective treatments (*p* ≤ 0.05).

**Figure 8 plants-13-00465-f008:**
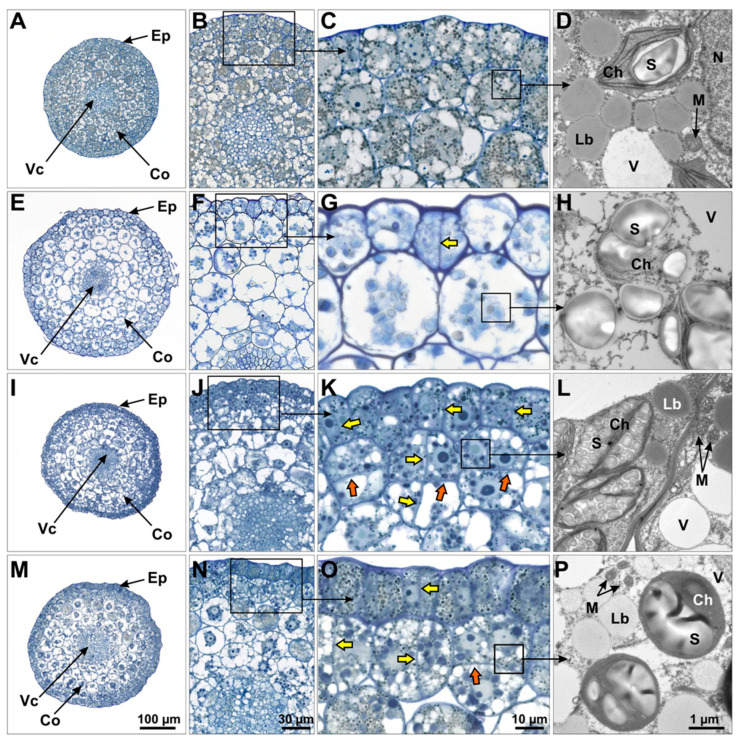
Light microscopy and transmission electron microscopy (TEM) analyses of cross-sectioned hypocotyls of immature embryos of cv. Ligena. Hypocotyl of freshly isolated embryo (**A**–**D**), after 4 days on culture medium without growth regulators (**E**–**H**), after 3 days on BAP-containing medium (**I**–**L**) and after 4 days on IAA-containing medium (**M**–**P**). After 4 days of culture, cells in the control have significantly expanded in size, displaying large vacuoles (**F**,**G**), with the cytoplasm typically containing starch granules, while lipid droplets are rare (**H**). Cells of hypocotyls cultivated on BAP- or IAA-containing medium hardly increase in size (**J**,**K**,**N**,**O**). Their vacuoles remain small and their cytoplasm accumulates many lipid droplets (**L**,**P**). The formation of new cell walls, revealing cell proliferation in epidermis and cortex, is indicated by colored arrows; anticlinal and periclinal divisions are indicated by yellow and orange arrows, respectively. Note that cell proliferation is strongly induced upon culture with IAA or BAP (**K**,**O**) compared to the control (**G**). The 3rd column shows enlarged images of boxed areas in the 2nd column. The 4th column shows enlarged TEM images of representative, boxed areas in the 3rd column. Scale bars, Ch, chloroplast; Co, cortex; Ep, epidermis; Lb, lipid bodies, M, mitochondrion; N, nucleus; S, starch; V, vacuole; Vc, vasculature.

**Figure 9 plants-13-00465-f009:**
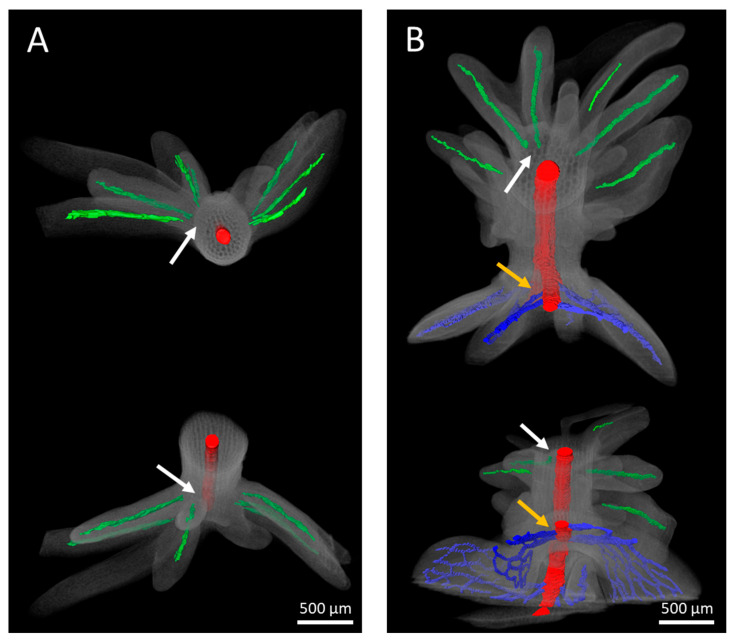
3-D reconstruction of developing leafy structures emerging at hypocotyls of immature embryos of cv. Ligena after two weeks on (**A**) BAP- or (**B**) IAA-containing induction media. The structures are each displayed from two different perspectives. Leafy structures feature unbranched veins (green) that are not connected to the central vasculature (red) of the cultivated zygotic embryo explant. White arrows indicate such missing connection sites, whereas the veins (blue) of the embryo’s cotyledons are connected to the central vasculature as indicated by orange arrows. This suggests that the veins present in the adventitious structures were formed newly and independently. These veins appear to have a starting point at the position of the former surface of the hypocotyl, which implies that their formation is initiated there and proceeds with the elongation of adventitious structures.

## Data Availability

Data are contained within the article and Appendix A.

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
