# Peer review of "Plant Regeneration via Adventitious Shoot Formation from Immature Zygotic Embryo Explants of Camelina"

_plants, 2024, doi:10.3390/plants13040465_

Round 1
Reviewer 1 Report (New Reviewer)
Comments and Suggestions for Authors
In this manuscript (plants-2839596) entitled "Plant regeneration via adventitious shoot formation from immature zygotic embryo explants of camelina" submitted to Plants, Barno Ruzimurodovna Rezaeva and colleagues have developed a plant regeneration method using immature embryos. This research is interesting and convincing, but minor points need to be addressed to improve the quality of this manuscript.
1. For Figure 1, at least three representative samples should be shown for each stage during formation of leafy structures at the hypocotyl of immature zygotic embryos (IZEs) of camelina in the revised Figure.
2. For Figure 2, number of leafy structures formed from immature zygotic embryos (IZEs) of camelina 212 cultivated on induction media without supplementation of IAA and BAP should be analyzed in the revision.
3. For Figure 3, Number of leafy structures formed from camelina IZEs cultivated on induction media supplemented with various concentrations (e.g. 0.2, 0.5, 1.0, 2.0 mg/l) of NAA, Dicamba, and 2.4 D should be analyzed in the revision.
4. For Figure 9, scale bars should be included in the revised figure.
Author Response
Authors' responses to the reviewers' comments
We are grateful to both reviewers for the points they have addressed. These comments have helped us a lot in identifying aspects that required corrections and improvements.
Reviewer 1
In this manuscript (plants-2839596) entitled "Plant regeneration via adventitious shoot formation from immature zygotic embryo explants of camelina" submitted to Plants, Barno Ruzimurodovna Rezaeva and colleagues have developed a plant regeneration method using immature embryos. This research is interesting and convincing, but minor points need to be addressed to improve the quality of this manuscript.
- For Figure 1, at least three representative samples should be shown for each stage during formation of leafy structures at the hypocotyl of immature zygotic embryos (IZEs) of camelina in the revised Figure.
Authors' response: It is rather unusual in research papers to show more than one representative example per developmental stage of such processes. In the present case of Figure 1, where there was only very little variability as to the developmental pattern among explants grown under the same conditions, images of more samples would actually not be more informative than the representative ones already shown. Moreover, the addition of more images per stage would render Figure 1 unnecessarily busy, and the many images would then have to be displayed smaller due to the limited space. Therefore, it would be more difficult to recognize the actually important information contained in this figure. We are kindly asking not to insist on the addition of more images. Since we do not have any further pictures of the relevant developmental stages available in comparable resolution and adequate quality, we would need to grow new donor plants from seeds and then cultivate another set of immature embryos. That is, the fulfillment of this request would delay the publication of this already conclusive study for more than four months. As a suitable alternative, we have additionally commented on Figure 1 in the text to better clarify the representativeness of the shown images.
- For Figure 2, number of leafy structures formed from immature zygotic embryos (IZEs) of camelina 212 cultivated on induction media without supplementation of IAA and BAP should be analyzed in the revision.
Authors' response: The response of IZEs cultivated on medium lacking any growth regulators is already comprehensively shown in Figure 8E through to H. Under this condition, no leafy structures were formed whatsoever. Having known this response prior to the experiment shown in Figure 2, we preferred to compare as many different concentrations of IAA and BAP as possible in such experiment rather than repeating a control without growth regulators.
- For Figure 3, Number of leafy structures formed from camelina IZEs cultivated on induction media supplemented with various concentrations (e.g. 0.2, 0.5, 1.0, 2.0 mg/l) of NAA, Dicamba, and 2.4 D should be analyzed in the revision.
Authors' response: In general, it is not possible to comprehensively optimize a new method of in vitro regeneration within the scope of a single study. Very many different media components and physical parameters could be varied for further improvement of such methods. The fine-tuning of the auxins we have preliminarily tested alongside IAA is one of these options indeed. In further experiments, however, we would rather investigate other factors with high priority that have not yet been varied at all. A particularly important approach would be, for example, to improve the conversion of the extraordinarily numerous leafy structures into shoots, for which auxins that are more active in the medium than IAA are, to the best of our experience with camelina, rather no good option. We had already pointed out the possibilities for future improvement in the manuscript. However, for the major conclusion and the novelty value of the present study, that immature camelina embryos are particularly suitable for the formation of adventitious shoots, further improvements of the method are not necessarily required in our opinion. The implementation of the experiment proposed by the reviewer would furthermore delay the publication of this already conclusive study by at least four months.
- For Figure 9, scale bars should be included in the revised figure.
Authors' response: In the revised manuscript, scale bars are included in the Figure 9.
Authors' responses to the reviewers' comments
We are grateful to both reviewers for the points they have addressed. These comments have helped us a lot in identifying aspects that required corrections and improvements.
Reviewer 1
In this manuscript (plants-2839596) entitled "Plant regeneration via adventitious shoot formation from immature zygotic embryo explants of camelina" submitted to Plants, Barno Ruzimurodovna Rezaeva and colleagues have developed a plant regeneration method using immature embryos. This research is interesting and convincing, but minor points need to be addressed to improve the quality of this manuscript.
- For Figure 1, at least three representative samples should be shown for each stage during formation of leafy structures at the hypocotyl of immature zygotic embryos (IZEs) of camelina in the revised Figure.
Authors' response: It is rather unusual in research papers to show more than one representative example per developmental stage of such processes. In the present case of Figure 1, where there was only very little variability as to the developmental pattern among explants grown under the same conditions, images of more samples would actually not be more informative than the representative ones already shown. Moreover, the addition of more images per stage would render Figure 1 unnecessarily busy, and the many images would then have to be displayed smaller due to the limited space. Therefore, it would be more difficult to recognize the actually important information contained in this figure. We are kindly asking not to insist on the addition of more images. Since we do not have any further pictures of the relevant developmental stages available in comparable resolution and adequate quality, we would need to grow new donor plants from seeds and then cultivate another set of immature embryos. That is, the fulfillment of this request would delay the publication of this already conclusive study for more than four months. As a suitable alternative, we have additionally commented on Figure 1 in the text to better clarify the representativeness of the shown images.
- For Figure 2, number of leafy structures formed from immature zygotic embryos (IZEs) of camelina 212 cultivated on induction media without supplementation of IAA and BAP should be analyzed in the revision.
Authors' response: The response of IZEs cultivated on medium lacking any growth regulators is already comprehensively shown in Figure 8E through to H. Under this condition, no leafy structures were formed whatsoever. Having known this response prior to the experiment shown in Figure 2, we preferred to compare as many different concentrations of IAA and BAP as possible in such experiment rather than repeating a control without growth regulators.
- For Figure 3, Number of leafy structures formed from camelina IZEs cultivated on induction media supplemented with various concentrations (e.g. 0.2, 0.5, 1.0, 2.0 mg/l) of NAA, Dicamba, and 2.4 D should be analyzed in the revision.
Authors' response: In general, it is not possible to comprehensively optimize a new method of in vitro regeneration within the scope of a single study. Very many different media components and physical parameters could be varied for further improvement of such methods. The fine-tuning of the auxins we have preliminarily tested alongside IAA is one of these options indeed. In further experiments, however, we would rather investigate other factors with high priority that have not yet been varied at all. A particularly important approach would be, for example, to improve the conversion of the extraordinarily numerous leafy structures into shoots, for which auxins that are more active in the medium than IAA are, to the best of our experience with camelina, rather no good option. We had already pointed out the possibilities for future improvement in the manuscript. However, for the major conclusion and the novelty value of the present study, that immature camelina embryos are particularly suitable for the formation of adventitious shoots, further improvements of the method are not necessarily required in our opinion. The implementation of the experiment proposed by the reviewer would furthermore delay the publication of this already conclusive study by at least four months.
- For Figure 9, scale bars should be included in the revised figure.
Authors' response: In the revised manuscript, scale bars are included in the Figure 9.

Reviewer 2 Report (New Reviewer)
Comments and Suggestions for Authors
The Authors presented a manuscript on plant regeneration from zygotic embryos in camelina. The paper is interesting and generally well written. Some minor comments are specified in pfd file.

Comments on the Quality of English LanguageEnglish is fine, needs minor corrections. Especially the term' silicles', is improper, and should be replaced by 'siliques' in whole paper.
Author Response
Authors' responses to the reviewers' comments
We are grateful to both reviewers for the points they have addressed. These comments have helped us a lot in identifying aspects that required corrections and improvements.
Reviewer 2
The Authors presented a manuscript on plant regeneration from zygotic embryos in camelina. The paper is interesting and generally well written. Some minor comments are specified in pfd file. In newer publications (2000 and later) this term is replaced by 'TORPEDO' stage. I suggest to use this term in whole paper. Or insert in th introduction an information about the nomenclature you used in the study for stages of embryo development including citation
Authors' response: The nomenclature of embryo developmental stages used in the manuscript is in accord with the latest comprehensive study we found on this topic (Feeney et al. 2018). This paper is cited in the Materials and Methods section where we also specified the lengths of the embryo size classes used in the experiments.
Hint as to Figure 2 (commented in the pdf)
Authors' response: The experiments described in Results subchapters 2.1.1 through 2.1.3 are confined to the scoring of leafy structure formation. Only as of subchapter 2.1.4., the approach was extended to the conversion of leafy structures to shoots and plants. We clearly discriminate leafy structures, shoots and rooted plants throughout the manuscript.
Hint as to Figure 8 (commented in the pdf): I suggest to enlarge this picture considerably, as this shows microscopic peculiarities, that has to be distinguish one from another.
Authors' response: We suggest displaying this figure in the paper in full page width. This is, however, to be decided by the journal's staff during the final processing.
Why siliques were stored in cold overnight? (commented in the pdf)
Authors' response: The collection of the silicles from the plants and their surface sterilization took quite some time. We stored the silicles over night to have a full working day for dissection and cultivation of a number of embryos as was required for conclusive experiments. In addition, we observed that this cold storage had a softening effect on the fruits, which facilitates dissection of the ovules.
if Authors used embryos differeing in size - did the siliques were in different age? (commented in the pdf)
Authors' response: Silicles were collected from the plants about 2 weeks after pollination without distinguishing any different fruit size classes. Only the embryos were then sorted according to their individual length during dissection.
Further comments and suggestions provided in the PDF file
Authors' response: We have considered all further minor comments and suggestions provided in the PDF file to improve the manuscript.
English is fine, needs minor corrections. Especially the term' silicles', is improper, and should be replaced by 'siliques' in whole paper.
Authors' response: Using the term silicle for camelina fruits is indeed perfectly in accord with the botanical nomenclature. Fruits are termed silicles instead of siliques when their length is less than three times their width. In other words, the silicle is a particularly short subtype of the silique.

This manuscript is a resubmission of an earlier submission. The following is a list of the peer review reports and author responses from that submission.
Round 1
Reviewer 1 Report
Comments and Suggestions for Authors
This manuscript deals with an in vitro plant regeneration method using immature zygotic embryos of the oilseed crop Camelina sativa. The method comprising adventitious shoot induction in hypocotyls of immature embryos was first established in the experimental line CAM139 and also applied to the commercial cultivar Ligena. Establishing in vitro regeneration protocols of valuable crops, such as camelina, is of interest in view of crop improvement by genetic manipulation including genome editing.
The regeneration results using immature zygotic embryos of camelina are quite innovative and the manuscript is well written. Nevertheless, it should be improved by adding more quantitative information on regenerated plants via shoots from hypocotyl leafy structures, at present shown by pictures in Fig. 6. The reported 80% in 2.1.4. referred to shoots forming roots should be better explained.
Some text from “Instructions” should be deleted at the end of 2.1.6. (L.185-187) and in Fig.1 legend.
Author Response
Responses to the reviewer (1) comments
We are grateful to both reviewers for the points they have addressed. These comments have helped us a lot in identifying aspects that required corrections or improvements in clarity.
Reviewer 1
The reported 80% in 2.1.4. referred to shoots forming roots should be better explained.
Authors' response: The rooting efficiency in 2.1.4. is now more comprehensibly explained and given by more concrete data.
Some text from “Instructions” should be deleted at the end of 2.1.6. (L.185-187) and in Fig.1 legend.
Authors' response: We apologize not to have spotted and deleted these internal comments earlier.

Reviewer 2 Report
Comments and Suggestions for Authors
The paper, on my opinion, does not represent a well evident innovative contribution, technically and scientifically relevant, on “Plant regeneration via adventitious shoot formation of camelina” in respect to previous studies on this topic also cited by the authors [8,9] and, in addition, the development of mature zygotic embryos of camelina was already shown by Pollard et al (2015). In conclusion, the innovative aspects are not well highlighted in the discussion which has to be highly improved from this point of view.
Concerning the statistical analysis, it must be reconsidered since in some figures, such as Fig. 3, Fig 4, letters denoting a statistically significant difference between their means, according to the Tukey test (P ≤ 0.05), are on columns having SE bars overlapping.
The abstract is too generic.
Comments on the Quality of English Languagegood enough
Author Response
Responses to the reviewer (2) comments
We are grateful to both reviewers for the points they have addressed. These comments have helped us a lot in identifying aspects that required corrections or improvements in clarity.
The paper, on my opinion, does not represent a well evident innovative contribution, technically and scientifically relevant, on “Plant regeneration via adventitious shoot formation of camelina” in respect to previous studies on this topic also cited by the authors [8,9].
Authors' response: The papers [8 and 9] of Tattersall and Millam (1999) and Yemets et al. (2013) do not meet the standards of publishing scientific results, since only anecdotal information is provided. Both papers contain claims that are not backed by data, i.e., no numbers of explants, no unambiguous data on responding explants, no data on rooting of regenerated shoots, and no data on the establishment of plants in soil are presented. Not to mention, that both papers are lacking experimental repetitions and statistical analyses. Therefore, these results cannot be seriously compared with what is presented in our manuscript. Moreover, with immature zygotic embryos, we have unprecedentedly used another type of explant source that features a particularly high level of totipotency. As a consequence, the pathway of regeneration in our method is, in contrast to the above papers, independent of callus formation, which might well be an advantage as to the avoidance of somaclonal and epigenetic variation. Admittedly, the approach presented here does certainly not constitute a methodological revolution, and yet are we convinced that our results should be made accessible to the scientific community.
The development of mature zygotic embryos of camelina was already shown by Pollard et al (2015).
Authors' response: Pollard et al. (2015) cultivated camelina embryos to investigate contents and fluxes of metabolites. They neither referred to any aspects of embryo development, nor did they provide data on developmental efficiency, viability, and capability of germination, not to mention adventitious shoot formation of these explants. Consequently, while agreeing that the Pollard study has for sure its scientific value, we do not consider this work as being relevant for our study and its aims.
Concerning the statistical analysis, it must be reconsidered since in some figures, such as Fig. 3, Fig 4, letters denoting a statistically significant difference between their means, according to the Tukey test (P ≤ 0.05), are on columns having SE bars overlapping.
Authors' response: This is a particularly important point. A critical revision of our data analyses has resulted in the identification of a true mistake in the previously applied procedure. Our data were now subjected to tests for normal (Gaussian) distribution and equal variance, both of which constitute essential preconditions for the applicability of the standard analysis of variance (ANOVA). Thereby, we identified experiments, where ANOVA was indeed inappropriately employed. In these cases, we instead applied the non-parametric (distribution-independent) Kruskal-Wallis analysis of variance on ranks followed by all pairwise comparison of the treatments using the Student-Newman-Keuls method, which has resulted in truely valid significances. Accordingly, in some revised diagrams, the data are now displayed as boxplots instead of means with standard deviations. Respective information is provided in the M&Ms section as well as in the respective figures and their captions.
In the previously submitted manuscript, we mistakenly used the term standard error of the means instead of standard deviation, with the latter, however, being displayed in the diagrams. Just of note, statistically significant differences may well be present even among treatments with overlapping standard deviations, since the statistical power of a test essentially depends on the scope of data, whereas the standard deviations are remaining largely the same in small as compared to even huge samples of a given population.
The abstract is too generic
Authors' response: Unfortunately, we do not quite get the point of this hint. While aim, concept and major experimental results of our study were already included in the abstract as clear as possible, we found and added just one essential aspect that was missing.

Round 2
Reviewer 2 Report
Comments and Suggestions for Authors
On my opinion the paper has not been enough improved and consequently it does not represent a contribution enough scientifically significant and original to be considered for publication on this journal
Author Response
Comment of Reviewer 2 on the 1st revision of the manuscript
On my opinion the paper has not been enough improved and consequently it does not represent a contribution enough scientifically significant and original to be considered for publication on this journal.
Authors' response
In this comment of Reviewer 2 on the 1st revision of our manuscript, key aspects of the manuscript, some of our responses to previous reviewer's comments and respective improvements made have not been taken into account. In the following, we list some points related to the manuscript's scientific significance and originality that in our opinion contradict the above comment of Reviewer 2. Unfortunately, it is unavoidable that there will be redundancies in the content of the following points with our previous responses to Reviewer 2's comments on the 1st version of the manuscript.
In the literature, the in vitro regeneration of camelina was reported only in anecdotal fashion, that is, without a solid data basis. In particular, no information on the experimental scope was provided and no data were presented as to the efficiency of plant regeneration. The two studies in question [8,9] are cited in the introduction of our manuscript, in the results section we provide the first comprehensive and conclusive experimental data on this subject, and in the discussion we have compared the novel principle used in our study with these conventional approaches.
Immature zygotic embryos have only rarely been used for adventitious shoot formation in dicotyledonous plant species, and our study is the very first to utilize this principle in camelina.
The use of immature zygotic embryos for the induction of adventitious shoots is not just any alternative among many, but has, thanks to the expected high totipotency of immature embryonic tissue, a particularly high potential for neoplastic and regenerative processes. Previously, this potential was not tapped on in camelina. In the present study, the expectations in this respect were confirmed by experimental results, which should also have a stimulating effect on the development of particularly suitable methods for other species such as rapeseed and other brassicas.
In contrast to previous reports on camelina, the regeneration of shoots and plants by our method usually takes place without an intermediate formation of callus, which can be of great advantage with regard to the genetic and epigenetic stability of the resulting plants.
The developmental processes underlying the established regeneration system, we described by means of unique microscopic analyses. These examinations have led to the identification of the tissue regions in which the induced adventitious shoots have their cellular origin. The fact that these are the outermost cell layers of the explants represents an extraordinarily favourable starting point for the intended application of this regeneration principle towards the development of methods for the transfer of recombinant DNA.
In addition, these microscopic examinations have shown that, contrary to the prevailing view to date, the formation of adventitious shoots does not necessarily involve a vascular connection between the initial explant and the adventitious shoots. This means that the formation of such vascular connections can no longer be used as a reliable criterion for distinguishing between somatic embryogenesis and adventitious shoot formation.
In our study, we succeeded in applying a method initially established using a model genotype directly and with sufficient efficiency to a current variety (cv. 'Ligena') for which no regeneration or transformation method had previously been available. Our own attempts of using the dipping method to transform 'Ligena' had been unsuccessful, despite this method's proven applicability to other Camelina accessions. Incidentally, this limitation was an important reason for dedicating ourselves to the innovative principle described in the manuscript.
